# MR image reconstruction using deep density priors

**Kerem C. Tezcan**[1]                                                    TEZCAN@VISION.EE.ETHZ.CH
**Christian F. Baumgartner**[1]                              BAUMGARTNER@VISION.EE.ETHZ.CH
**Roger Luechinger**[2]                                       LUECHINGER@BIOMED.EE.ETHZ.CH
**Klaas P. Pruessmann**[2]                                 PRUESSMANN@BIOMED.EE.ETHZ.CH
**Ender Konukoglu**[1]                                     ENDER.KONUKOGLU@VISION.EE.ETHZ.CH

[1] *Computer Vision Lab, ETH Zürich, Switzerland*

[2] *Institute for Biomedical Engineering, ETH Zürich and University of Zürich, Switzerland*

**Editors:** Under Review for MIDL 2019

## Abstract

We present a recently published work on undersampled MR image reconstruction (Tezcan et al., 2018) relying on deep learning (DL). The method uses a variational autoencoder trained on fully sampled images as the prior in a maximum a posteriori formulation of the reconstruction problem. Doing this allows decoupling the prior from the encoding, i.e. undersampling scheme and coil setting,allowing using the same network with any encoding without retraining, an aspect not guaranteed for any other reconstruction method using DL. Results indicate highly competitive performance.

**Keywords:** MRI, image reconstruction, density estimation, VAE.

## 1. Introduction

Deep learning (DL) provides a framework for extracting information from existing datasets, which can be used as prior information for completing missing k-space data in undersampled magnetic resonance (MR) image reconstruction. Recently proposed DL based methods implement this idea by learning a mapping from the undersampled image to a fully sampled image using corresponding image pairs during training. This mapping is either directly used to transform undersampled images (Kwon et al., 2017; Lee et al., 2017) or it is combined with an additional term ensuring data consistency (Schlemper et al., 2017). In either case the learned mapping is specific to the undersampling scheme and factor that were used in the training. We have shown in (Tezcan et al., 2018) that with this approach, if the undersampling scheme differs at test time from the training scheme, the reconstruction quality drops.

In this abstract we present our recently published method for MR image reconstruction using DL. In contrast to the previous approaches, we propose training a variational autoencoder (VAE) to model the distribution of fully sampled images. This distribution is then used as the prior in the maximum a posteriori (MAP) estimation of the image. Such a prior has the advantage of being decoupled from the encoding operation, i.e. independent of coil settings and undersampling schemes. In this abstract we present the method and some key results and refer the interested readers to the main article for the details.

Table 1: Table showing the mean (and standard deviation) of different reconstruction quality metrics. Top group: full FOV, bottom group: cropped FOV.

| | R=2 | | R=3 | | R=4 | | R=5 | |
|---|---|---|---|---|---|---|---|---|
| | RMSE | CNR | RMSE | CNR | RMSE | CNR | RMSE | CNR |
| FS | - | 0.48(0.10) | - | 0.48(0.10) | - | 0.48(0.10) | - | 0.48(0.10) |
| Zero filled | 13.03(1.13) | 0.40(0.09) | 21.15(1.36) | 0.33(0.07) | 24.92(1.91) | 0.31(0.06) | 27.36(1.79) | 0.30(0.06) |
| DDP | 2.76(0.53) | 0.48(0.11) | 4.25(0.61) | **0.48**(0.10) | **6.46**(1.57) | **0.46**(0.11) | **11.13**(2.39) | **0.41**(0.10) |
| TV (Rudin et al., 1992) | 3.87(0.47) | 0.46(0.11) | 7.56(0.83) | 0.40(0.10) | 11.40(1.39) | 0.35(0.09) | 14.56(1.23) | 0.31(0.08) |
| DLMRI (Ravishankar and Bresler, 2010) | 4.48(0.52) | 0.46(0.11) | 7.25(0.83) | 0.40(0.10) | 10.72(1.31) | 0.33(0.09) | 13.87(1.25) | 0.30(0.08) |
| ADMMNet (Yang et al., 2016) | 3.55(0.40) | 0.48(0.11) | 7.06(0.52) | 0.45(0.11) | 11.26(0.72) | 0.36(0.09) | 13.05(0.70) | 0.32(0.08) |
| BM3D-MRI (Dabov et al., 2007) | **1.92**(0.36) | 0.48(0.10) | **4.23**(1.05) | 0.46(0.10) | 8.08(2.48) | 0.43(0.10) | 11.70(2.76) | 0.38(0.09) |
| DDP | 2.68(0.38) | **0.48**(0.10) | 4.61(1.12) | **0.47**(0.10) | 7.39(1.47) | **0.45**(0.10) | 13.00(3.01) | **0.39**(0.08) |
| SIDWT (Ning et al., 2013) | 4.49(0.98) | 0.45(0.11) | 9.42(1.62) | 0.39(0.09) | 14.57(1.96) | 0.33(0.08) | 18.76(2.80) | 0.32(0.07) |
| FDLCP (Zhan et al., 2016) | **2.63**(0.35) | **0.48**(0.10) | **4.35**(0.87) | 0.45(0.10) | **6.72**(0.89) | 0.41(0.10) | **9.62**(1.48) | 0.35(0.08) |
| PBDW (Qu et al., 2012) | 3.24(0.38) | 0.47(0.11) | 5.59(0.94) | 0.44(0.10) | 8.51(0.98) | 0.38(0.09) | 11.38(1.39) | 0.34(0.08) |

## 2. Method

### 2.1. The MAP formulation of reconstruction

We model the acquisition as $y = Em + \eta_\sigma$, where $y \in C^{N\gamma}$ is the undersampled k-space data, $E$ the undersampled encoding operation with $\gamma$ coils, $m \in C^M$ the image with $M > N$ and $\eta_\sigma$ normally distributed complex noise with standard deviation $\sigma$. The MAP estimation problem after the log transformation is written as $\arg\max_m \log p(m|y) = \arg\max_m \log p(y|m) + \log p(m)$, where $p(m)$ is the prior on the images. In this work, we propose to approximate it with a VAE.

### 2.2. Learning the prior with VAEs

The VAE (Kingma and Welling, 2013; Rezende et al., 2014) is a latent variable model used for approximating a probability distribution $p(x)$ provided a training set containing enough samples $\{x_i\}$ from the distribution. It operates by maximizing a lower bound called the evidence lower bound (ELBO) to the target $p(x)$. The ELBO is defined using neural networks, whose parameters are learned in the training process, so that the ELBO approximates the distribution $p(x)$. In the scope of this work, we train the VAE with 28x28 magnitude image patches $\{|x|_i\}$ from the training images. For reconstruction we use the facts that i) once the VAE is trained $p(|x|) \approx ELBO(|x|)$ and ii) the ELBO is differentiable according to $|x|$.

### 2.3. Reconstruction using the ELBO

We use an algorithm based on projection onto convex sets (POCS) to obtain the MAP estimate (De Pierro and Helou Neto, 2009). We use a projection $P_{DC}$ for the data likelihood term replacing the sampled k-space positions with the measured values and doing coil combination. This corresponds to POCS-SENSE (Samsonov et al., 2004) with multiple coils. To do the prior projection $P_{prior}$ on the image $m$ we solve a sub-maximization problem for a set of overlapping patches in the image, where we do K gradient ascent steps to maximize the ELBO for each patch as $|x| \leftarrow |x| + \alpha \frac{dELBO(|x|)}{d|x|}$. The prior acts only on the magnitude, hence the method requires a separate procedure to correct the phase. Given multiple coils, POCS-SENSE reconstruction can achieve this, otherwise we use an additional projection for the phase $P_{phase}$, which enforces smooth phase images. The full reconstruction scheme

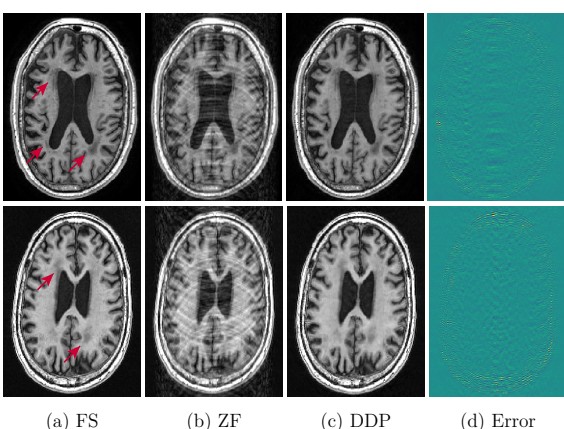

(a) FS      (b) ZF      (c) DDP      (d) Error

Figure 1: Reconstructions for two ADNI images, as well as the fully sampled (FS) and zero-filled (ZF) images. (Error maps clipped to $\pm0.3$)

(a) FS      (b) ZF      (c) DDP      (d) Error

Figure 2: Reconstruction results for acquired data (error maps clipped to $\pm0.3$) using ESPIRiT coil maps for R=2 Cartesian and R=4 radial US patterns (lower row). In both cases the prior projection is the same, only the data consistency projection differs (using FFT and NUFFT, respectively).

for the proposed deep density prior (DDP) method can then be written as

$$m^{t+1} = P_{DC}P_{phase}P_{prior}m^t. \tag{1}$$

After $T$ iterations we obtain the image $m^T$ as the MAP estimate. We use the zero-filled image for $m^0$, two sets of overlapping patches, $\alpha = 10^{-4}$, K=10 and T=30.

## 3. Experiments and Results

We used 790 T1 weighted (T1w) central slices extracted from 158 subjects (5 from each) from the HCP dataset (Van Essen et al., 2013) to train the VAE. We then evaluated the methods with 17 central slices each from different test subjects. We acquired raw k-space brain images from 8 subjects with 16 coils (similar acquisition parameters to HCP). We also used two T1w images with lesions from the ADNI dataset (http://adni.loni.usc.edu/). We retrospectively undersampled with factors R and reconstructed the images using Cartesian (with 15 fully sampled central profiles) and radial patterns. We used ESPIRiT (Uecker et al., 2014) to obtain the coil maps for the acquired data.

We report normalized root mean squared error (RMSE) values in percentage and contrast-to-noise ratio (CNR) values of multiple methods from the literature in Table 1 for comparison purposes. We do two sets of experiments, one with full and one with cropped FOVs, since some methods work only with square FOVs. The proposed method yields the best results for the full FOV case for $R > 2$ in terms of RMSE and the best CNR values for all factors. The results are also competitive for the cropped FOV. These numbers indicate the method is capable of reconstructing details without introducing unwanted smoothness.

We show examples of reconstructed images in Figure 1 from the ADNI dataset. In Figure 2 we show reconstruction results from the retrospectively undersampled k-space data, where we used NUFFT (Lin and Chung, 2017) in the encoding operation for the radial sampling. The method can faithfully reconstruct the lesions and yields good visual quality in all cases.

## Acknowledgments

This work was supported by Swiss National Science Foundation (grant number: 205321_173016). We thank Nvidia for their GPU donation. We used multiple images from the HCP dataset (https://www.humanconnectome.org/) and two images from the ADNI dataset, see http://adni.loni.usc.edu/wp-content/uploads/how_to_apply/ADNI_Acknowledgement_List.pdf for a comprehensive list of involved parties.

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
