# OpenReview forum: "MR image reconstruction using deep density priors"
_MIDL.io/2019/Conference/Abstract — MIDL Abstract 2019_

### Official Review · AnonReviewer2 · 2019-04-29
**variational auto-encoder captures priors invariant to encoding, undersampling scheme and coil settings**

**Rating:** 4
**Confidence:** 2

**Review:**

The paper proposes MRI reconstruction via DL and in particular using a variational auto encoder to capture the prior distribution of the data. The approach is well developed, it has been compared with a number of state of the art methods and baseline approaches,  has been trained and tested on sufficient amount of data and is interesting all around.

The paper has actually already been published on TMI, therefore it's not novel. this doesn't violate the guidelines of MIDL for abstract presentation and therefore this doesn't constitute an issue.

---

### Official Review · AnonReviewer1 · 2019-05-01
**accept**

**Rating:** 3
**Confidence:** 2

**Review:**

Overall, the abstract describes the problem, the solution and indicates good empirical results. It is a clinically relevant problem and the proposed solution is sensible and may deserve to be presented at the conference.

---

### Decision · Program_Chairs · 2019-05-06
**Acceptance Decision**

Accept